# Trends of non-vaccination, under-vaccination and missed opportunities for vaccination (2003–2014) amongst children 0–23 months in Kenya

**Christopher Ochieng' Odero**[1]*, **Doreen Othero**[1], **Vincent Omondi Were**[2], **Collins Ouma**[3]

1 Department of Public Health, Maseno University, Kisumu, Kenya, 2 KEMRI Wellcome-Trust Research Program, Health Economics Research Unit, Kilifi, Kenya, 3 Department of Biomedical Sciences and Technology, Maseno University, Kisumu, Kenya

* christodero@gmail.com

**Data Availability Statement:** All data and related metadata underlying the findings reported have

## Abstract

Vaccines are effective and cost-effective. Non-vaccination, under-vaccination, and missed opportunities for vaccination (MOV), have contributed to incomplete vaccination coverage in Kenya. Analyzing their trends is essential for targeting interventions and improvement strategies. This study aimed to assess trends of non-vaccination, under-vaccination, and MOV among children aged 0–23 months in Kenya using data obtained from the Kenya Demographic and Health Surveys (KDHS) conducted in 2003, 2008/09, and 2014. A two-stage, multi-stage, and stratified sampling technique was used. Weighted analysis was conducted to ensure generalizability to the full population. Using the KDHS sample size estimation process, the sample size was estimated for each indicator, with varying standard error estimates, level of coverage and estimated response rates. Final sample size was 2380 (2003), 2237 (2008/09) and 7380 (2014). To determine the level of non-vaccination, under-vaccination and MOV among children aged 0–23 months, a weighted descriptive analysis was used to estimate their prevalence, with 95% confidence intervals (CI) for each year. MOV was defined using an algorithm as a binary variable. Data coding and recoding were done using Stata (version 14; College Station, TX: StataCorp LP). Trends in proportions of non-vaccination, under-vaccination and MOV were compared between 2003, 2008/09, and 2014 using the Cochrane-Armitage trend test. All results with $P \leq 0.05$ were considered statistically significant. Trends in proportion of non-vaccination among children aged 0–23 months in Kenya was 13.2%, 6.1% and 3.2% in 2003, 2008/09 and 2014, respectively ($P = 0.0001$). Trends in proportion of under-vaccination among children aged 0–23 months in Kenya was 54.3%, 50% and 51.3% in 2003, 2008/09 and 2014, respectively ($P = 0.0109$). The trends in proportion of children who experienced MOV was 22.7% in 2003, 31.9% in 2008/09 and 37.6% in 2014 ($P = 0.0001$). In the study duration, non-vaccination decreased by 10%, under-vaccination remained relatively stable, and MOV increased by ~15%. There is need for the Government and partners to implement initiatives that improve vaccine access and

been provided as part of the submitted article. The datasets used in this analysis were obtained from the DHS program webpage (https://dhsprogram. com). The relevant datasets regarding the children's information (Kids recode [KR]) were used. The STATA format were extracted and analyzed. We confirm that others would be able to access or request these data in the same manner as the authors.

**Funding:** The authors received no specific funding for this work.

**Competing interests:** The authors have declared that no competing interests exist.

coverage, particularly in regions with low coverage rates, and to address missed opportunities for vaccination.

## Introduction

Vaccines play a vital role in preventing and controlling infectious diseases, making a significant contribution to global public health [1–3]. Despite their proven effectiveness, achieving optimal vaccination coverage remains challenging, particularly in low- and middle-income countries like Kenya [4]. This study aims to examine patterns of non-vaccination, under-vaccination, and Missed Opportunities for Vaccination (MOV) among children aged 0–23 months in Kenya from 2003 to 2014.

The success of immunization programs relies on factors such as vaccination coverage, reporting accuracy, and strategies to reach every eligible child [5]. Failing to achieve complete vaccination coverage can be attributed to non-vaccination, under-vaccination, and missed opportunities for vaccination, emerging as significant public health concerns [6,7].

In Africa, millions of children suffer from vaccine-preventable diseases (VPDs) due to limited access to immunization services, resulting in hundreds of thousands of deaths annually [8]. Specifically, Kenya has faced challenges in meeting national vaccination targets, with the proportion of fully immunized children consistently below the desired 80% threshold [9].

Non-vaccination occurs when eligible children receive zero vaccine doses, posing an increased risk of disease transmission [10]. Common reasons for non-vaccination include lack of knowledge, religious taboos, complacency, and logistical challenges [10,11]. Under-vaccination involves incomplete immunization schedules and is associated with VPD outbreaks globally [12].

Missed Opportunities for Vaccination (MOV) happen when eligible children interact with health services but do not receive all eligible vaccine doses [13]. The prevalence of MOV in Africa ranges from 27–32%, highlighting the need for targeted interventions [14,15].

This study addresses gaps in understanding the trends of non-vaccination, under-vaccination, and MOV in Kenya. Existing data is fragmented and unreliable, hindering comprehensive assessments. Additionally, systematic evaluations of intervention programs and a focus on vaccine equity, particularly among marginalized populations, are lacking. These gaps underscore the necessity for evidence-based interventions and further research to enhance vaccination coverage and equity in Kenya. The findings will inform policymakers and contribute to a better understanding of the drivers of vaccine confidence in the Kenyan context. The study utilizes nationally representative data from 2003 to 2014 to provide a thorough analysis of vaccination trends in the specified age group.

## Materials and methods

### Study design

This was a quantitative study that analyzed secondary data from longitudinal repeated annual and national cross-sectional surveys obtained from the Kenya demographic and health surveys (KDHS) conducted in 2003, 2008/9, and 2014. The datasets are publicly available at https:// dhsprogram.com. To capture fundamental child health changes and policies related to vaccinations that have occurred over the last years, a 10-year trend analysis offered better insight into the possible trends in vaccination gaps over time.

## Study area

This study covered the entire 47 counties in Kenya. The KDHS surveys were national level surveys, which in 2003 and 2008/09 were conducted across all the former eight Provinces of Kenya and in 2014, was conducted in all the 47 counties in line with the new administrative units.

## Study population

The study population were children aged 0–23 months for each year of the KDHS studies.

## Sampling technique

The KDHS used a two-stage, multi-stage and stratified technique with households as the sampling unit. Within each household, eligible women 15–49 years were interviewed. The KDHS questionnaires included a mother questionnaire that has both mother and child information. The sampling weights were created to account for the uneven distribution of sampling probabilities and non-responses rates. These allowed for a weighted analysis to account for full population generalizability. In this analysis, full child datasets for years 2003, 2008/9 and 2014 were used.

Sampling weights were formulated to rectify variations in the likelihood of selecting sampling units, households, and individuals, while accounting for the response rate per stratum, which encompassed counties and enumeration areas. The household weight for a specific household was computed as the reciprocal of its household selection probability, multiplied by the reciprocal of the household response rate within the stratum. Subsequently, the individual weight was derived by multiplying the household weight by the reciprocal of the individual response rate within the same stratum. Given the pivotal concept of response rate for groups in the KDHS, households and individuals were categorized into sample strata to calculate response rates. This approach ensured that the analysis encompassed a representative sample of the population, and upheld the statistical integrity of the findings.

## Sample size

The study used the KDHS sample size estimation process, which aimed to estimate the minimum number of women aged 15–49 years, number of households, number of children under five years and 12–23 months. In addition, in 2014, the sample size calculation was made to account for county level estimates in line with the devolved units. The sample size was estimated for each indicator, with varying standard error estimates, level of coverage and estimated response rates [16]. Whilst the sample size for full immunization coverage of children aged 12–23 months was used in the determination of non-vaccination and under-vaccination, this study was restricted to children aged 0–23 months for all the years. The sample size for full immunization coverage of children aged 0–23 months in 2014 was as follows;

$$ n = Deft^2 \frac{\left(\frac{1}{p} - 1\right)}{\alpha^2} / (R_{i\,X}\, R_{h\,X}\, d) $$

where;

n–The sample size in households;
Deft–The design effect of 1.800
P–The estimated proportion (0.792)
$\alpha$ –the desired relative standard error; (SE = 0.011)
$R_i$–the individual response rate; 92.6% (0.926)

$R_h$–the household gross response rate; 98% (0.98)

d–the number of eligible individuals per household.1.05

Based on these assumptions, the sample size was 2380 in 2003, 2237 in 2008/2009 and 7380 in 2014.

### Data management and statistical analysis

The DHS data is publicly available and was obtained from the DHS program webpage (https://dhsprogram.com) [17] for the period 2003, 2008/9, and 2014. The relevant datasets regarding the children's information (Kids recode [KR]) were used. The STATA format were extracted and analyzed. All analysis datasets and all reports generated from this data were stored in an access-controlled google drive, only accessible to the investigator. All coding and recoding's were done using Stata version 14.

To determine the level of non-vaccination, under-vaccination and missed opportunities for vaccination among children aged 0–23 months, a weighted descriptive analysis was used to estimate the prevalence of non-vaccination and under-vaccination, with 95% confidence intervals (CI) for each year. The MOV was defined using an algorithm as a binary variable. The outcome variable was categorized as MOV = 1 or 0 otherwise. The Cochrane-Armitage trend test was used to compare trends in proportions of non-vaccination, under-vaccination and missed opportunities for vaccination between 2003, 2008/9, and 2014. For all analyses, p≤0.05 was considered statistically significant.

### Ethics approval and consent

Ethical review and approval was sought from the Maseno University Institutional Review Board/ Ethics Review Committee (IRB/ERC) before the start of study procedures. The team further sought research permit from the National Commission for Science, Technology and Innovations (NACOSTI). No informed consent was obtained before the secondary analysis. Consent provided during the KDHS typically includes a statement that allows for the data collected to be used for secondary analysis. To protect the confidentiality of the participants and their data, this secondary analysis used anonymized data.

## Results

### Trends of non-vaccination amongst children 0–23 months in Kenya 2003 to 2014

In the three surveys, the percentage of children who were non-vaccinated decreased significantly from 13.2% (2003) to 6.1% (2008/09) to 3.2% (2014); $P<0.0001$ (**Table 1**). In terms of residence, a notable decrease in non-vaccination trends was observed among children from the rural areas with a decrease from 14.7% (2003) to 6.6% (2008/09) to 4.0% (2014); $P<0.0001$. This pattern was observed for all the significant results across all the factors except for women who were divorced/ separated or widowed 12.0% (2003) to 3.2% (2008/09) and 3.9% (2014); $P = 0.0001$ and for women residing in Nairobi Province 4.6% (2003) to 6.1% (2008/09) and 1.0% (2014); $P = 0.0015$, respectively.

### Trends of under-vaccination among children aged 0–23 months in Kenya from 2003 to 2014

The trend of under-vaccination from 2003 to 2008/09 to 2014 showed varied results as displayed in **Table 2**. It decreased from 54.3% (2003) to 50.0% (2008/09) to 51.3% (2014); $P = 0.0109$. Among female children, under-vaccination decreased from 54.7% (2003) to 48.3%

**Table 1. Trend of non-vaccination among children aged 0–23 months in Kenya; KDHS 2003, 2008/09 and 2014.**

| | 2003 (n = 2380) | 2008/09 (n = 2237) | 2014 (n = 7380) | P-value[a] |
|---|---|---|---|---|
| | % (95% CI) | % (95% CI) | % (95% CI) | |
| **Sex of child** | | | | |
| Male | 13.5(11.7–15.6) | 6.8(5.4–8.3) | 2.7(2.1–3.2) | <0.0001 |
| Female | 12.9(11.1–14.9) | 5.5(4.3–7.0) | 3.8(3.2–4.5) | <0.0001 |
| **Residence** | | | | |
| Urban | 6.9(4.9–9.7) | 4.4(2.8–6.7) | 1.8(1.3–2.4) | <0.0001 |
| Rural | 14.7(13.1–16.3) | 6.6(5.5–7.8) | 4.0(3.5–4.6) | <0.0001 |
| **Mother's Age** | | | | |
| 15–19 | 12.1(8.8–16.5) | 6.3(3.7–10.5) | 3.7(2.5–5.4) | <0.0001 |
| 20–24 | 13.1(10.8–15.7) | 5.0(3.6–6.9) | 2.5(1.9–3.2) | <0.0001 |
| 25–29 | 11.7(9.3–14.5) | 5.8(4.1–8.0) | 2.9(2.3–3.7) | <0.0001 |
| 30–34 | 13.0 (10.1–16.6) | 7.2(5.1–10.1) | 2.6(1.9–3.6) | <0.0001 |
| 35–39 | 17.4(13.0–22.8) | 9.7(6.3–14.7) | 3.6(2.5–5.1) | <0.0001 |
| 40–44 | 13.3(7.8–21.8) | 4.5(1.6–11.8) | 12.7(9.2–17.3) | 0.8814 |
| 45–49 | 38.2(17.9–63.7) | 4.0(0.3–34.9) | 3.5(0.65–16.8) | 0.0006 |
| **Marital Status** | | | | |
| Never Married | 9.5(6.0–14.6) | 5.8(3.4–9.7) | 2.5(1.5–4.0) | <0.0001 |
| Married/ living together | 13.7(12.2–15.2) | 6.4(5.4–7.7) | 3.2(2.8–3.7) | <0.0001 |
| Divorced/ separated/ widowed | 12.0(7.9–17.8) | 3.2(1.4–7.4) | 3.9(2.6–5.9) | 0.0001 |
| **Religion** | | | | |
| Roman catholic | 12.4(10.0–15.2) | 5.8(4.0–8.3) | 2.0(1.3–2.9) | <0.0001 |
| Protestant/ other Christian | 11.6(10.1–13.4) | 5.2(4.1–6.4) | 3.0(2.6–3.5) | <0.0001 |
| Muslim | 23.1(17.8–29.3) | 10.9(7.2–16.0) | 7.1(5.3–9.4) | <0.0001 |
| No religion | 22.7(14.1–34.4) | 13.7(7.5–23.6) | 4.8(2.6–8.8) | <0.0001 |
| **Birth Order** | | | | |
| 1 | 10.2(7.9–12.9) | 4.0(2.6–6.1) | 2.3(1.7–3.1) | <0.0001 |
| 2–4 | 10.9(9.2–12.8) | 4.6(3.5–6.0) | 2.4(1.9–2.9) | <0.0001 |
| 5 + | 19.9(17.0–23.1) | 10.8(8.5–13.5) | 6.4(5.3–7.7) | <0.0001 |
| **Parity** | | | | |
| 0–1 | 11.5(9.2–14.3) | 4.0(2.6–6.0) | 2.1(1.6–2.9) | <0.0001 |
| 2–4 | 10.4(8.8–12.2) | 5.0(3.9–6.4) | 2.5(2.1–3.1) | <0.0001 |
| 5 + | 21.2(18.0–24.8) | 10.9(8.5–13.9) | 6.7(5.5–8.1) | <0.0001 |
| **Number of Children in Household** | | | | |
| 0–1 | 11.3(9.3–13.6) | 3.0(2.0–4.3) | 1.9(1.5–2.5) | <0.0001 |
| 2–4 | 14.3(12.6–16.2) | 7.9(6.6–9.4) | 4.2(3.6–4.8) | <0.0001 |
| 5 + | 10.9(2.9–33.7) | 4.9(1.3–17.1) | 1.8(2.8–11.3) | 0.0738 |
| **Education** | | | | |
| No Education | 29.6(25.1–34.6) | 11.3(8.1–15.6) | 8.9(7.1–11.0) | <0.0001 |
| Primary Incomplete | 15.0(12.7–17.5) | 6.6(5.0–8.7) | 3.6(2.9–4.5) | <0.0001 |
| Primary Complete | 7.8(5.9–10.1) | 5.5(4.0–7.4) | 2.3(1.7–3.0) | <0.0001 |
| Secondary + | 5.5(3.8–7.8) | 3.8(2.5–5.8) | 1.7(1.3–2.3) | <0.0001 |
| **Wealth Quintile** | | | | |
| Lowest | 25.0(21.6–28.6) | 11.2(8.8–14.2) | 6.6(5.5–7.8) | <0.0001 |
| Second | 12.3(9.7–15.5) | 4.3(2.8–6.7) | 2.5(1.8–3.5) | <0.0001 |
| Middle | 11.0(8.4–14.2) | 7.5(5.3–10.5) | 3.5(2.7–4.7) | <0.0001 |
| Fourth | 6.9(4.8–9.8) | 3.4(2.0–5.7) | 1.4(0.8–2.2) | <0.0001 |
| Highest | 6.6(4.6–9.3) | 3.0(1.8–5.1) | 1.0(0.6–1.6) | <0.0001 |

*(Continued)*

**Table 1.** (Continued)

| | 2003 (n = 2380) | 2008/09 (n = 2237) | 2014 (n = 7380) | P-value[a] |
|---|---|---|---|---|
| | % (95% CI) | % (95% CI) | % (95% CI) | |
| **Occupation** | | | | |
| Unemployed | 14.7(12.5–17.1) | 7.2(5.7–9.0) | 3.7(2.8–4.8) | <**0.0001** |
| Employed | 12.3(10.7–14.1) | 5.4(4.3–6.8) | 2.1(1.6–2.8) | <**0.0001** |
| **Province** | | | | |
| Nairobi | 4.6(2.2–9.4) | 6.1(3.1–11.9) | 1.0(0.4–2.0) | **0.0015** |
| Central | 6.3(3.9–10.1) | 6.1(3.3–11.1) | 0.4(0.1–1.4) | <**0.0001** |
| Coast | 9.2(6.0–13.9) | 5.2(2.9–9.1) | 2.3(1.4–3.5) | <**0.0001** |
| Eastern | 676.4(4.3–9.4) | 4.1(2.3–6.8) | 1.1(0.5–2.0) | <**0.0001** |
| Nyanza | 25.0(20.7–29.7) | 7.2(5.1–10.0) | 2.6(1.8–3.8) | <**0.0001** |
| Rift Valley | 11.9(9.8–14.6) | 4.8(3.3–6.7) | 4.7(3.9–5.7) | <**0.0001** |
| Western | 14.3(10.8–18.7) | 7.5(4.8–11.5) | 4.2(3.0–5.7) | <**0.0001** |
| North Eastern | 56.7(44.2–68.4) | 21.9(13.4–33.7) | 15.2(11.1–20.4) | <**0.0001** |
| **Place of delivery** | | | | |
| Home | 18.1(16.2–20.2) | 8.5(7.1–10.3) | 6.6(5.7–7.7) | <**0.0001** |
| Public | 5.5(3.8–7.6) | 2.9(1.9–4.3) | 1.5(1.1–1.9) | <**0.0001** |
| Private | 6.4(4.2–9.6) | 5.1(2.9–8.6) | 1.1(0.7–1.9) | <**0.0001** |
| **Total** | 13.2(11.9–14.6) | 6.1(5.2–7.2) | 3.2(2.8–3.6) | <**0.0001** |

KDHS = Kenya Demographic Health Survey.

[a] P-values were calculated form the Cochrane-Armitage trend test.

(2008/09) then increased to 50.8% (2014); $P = 0.0194$. For those residing in the urban areas, under-vaccination decreased from 58.3% (2003) to 43.3% (2008/09) and then increased to 49.4% (2014); $P = 0.0005$. The percentages of children born to mothers aged 25–29 years was highest in 2003 (54.3%) then 2008/09 (51.6%) to 2014 (48.1%); $P = 0.0073$, and for women who were divorced/ separated/ widowed 2003 (58.2%), 2008/09 (56.8%) and 2014 (46.6%); $P = 0.0077$.

In terms of birth order and parity, under-vaccination among children who were second to forth born decreased from 55.9% (2003) to 51.1% (2008/09) to 51.0% (2014); $P = 0.0035$ and 56.5% (2003) to 51.5% (2008/09) and 50.7% (2014); $P = 0.0004$ respectively. Under-vaccination amongst women having zero or one child living in a household decreased from 51.7% (2003) to 42.1% (2008/09) then increased to 46.7% (2014); $P = 0.0097$. For children born to mothers in the lowest wealth quintile it was 52.0% (2003) to 52.6% (2008/09) and then increased to 57.1% (2014); $P = 0.0299$. Amongst those in second level wealth quintile, under-vaccination decreased from 59.1% (2003) to 55.2% (2008/09) and to 52.1% (2014); $P = 0.0070$). In those within the fourth level of wealth quintile, under-vaccination decreased from 56.1% (2003) to 49.3% (2008/09) and then to 47.2% (2014); $P = 0.0036$. Amongst those in the highest wealth quantile, under-vaccination was 54.7% (2003) then reduced to 43.1% (2008/09) and then rose to 47.2% (2014); $P = 0.0056$.

In terms of occupation, trends in under-vaccination among children whose mothers were employed decreased from 53.6% (2003) to 49.5% (2008/09) and then increased to 50.1% (2014); $P = 0.0372$. In the Provinces, the trend of under-vaccination was also varied as shown in Western and North-Eastern Provinces (NEP). In Western, under-vaccination was 56.6% (2003), 51.9% (2008/09) and 49.9% (2014); $P = 0.0441$ while in NEP, it was 36.5% (2003), 49.7% (2008/09) and 50.6%; $P = 0.0468$). Additionally, trends in under-vaccination among

**Table 2. Trends of Under-vaccination among children aged 0–23 months in Kenya; KDHS 2003, 2008/09 and 2014.**

| | 2003 (n = 2380) | 2008/09 (n = 2237) | 2014 (n = 7380) | P-value[a] |
|---|---|---|---|---|
| | % (95% CI) | % (95% CI) | % (95% CI) | |
| **Sex of child** | | | | |
| Male | 53.9(51.0–56.7) | 51.6(48.7–54.5) | 51.8(51.8–53.4) | 0.2067 |
| Female | 54.7(54.3–57.5) | 48.3(45.3–51.3) | 50.8(49.2–52.5) | **0.0194** |
| **Residence** | | | | |
| Urban | 58.3(53.7–62.8) | 43.3(38.8–47.9) | 49.4(47.5–51.3) | **0.0005** |
| Rural | 53.4(51.1–55.6) | 51.7(49.4–54.0) | 52.3(50.9–53.7) | 0.4136 |
| **Mother's Age** | | | | |
| 15–19 | 62.1(56.3–67.6) | 54.4(47.6–61.0) | 60.6(56.9–64.1) | 0.6620 |
| 20–24 | 53.7(50.1–57.2) | 53.9(50.2–57.5) | 51.6(49.5–53.7) | 0.3232 |
| 25–29 | 54.3(52.7–58.2) | 51.6(47.6–55.6) | 48.1(46.0–50.2) | **0.0073** |
| 30–34 | 54.0(49.1–58.7) | 41.2(36.6–45.9) | 49.9(47.1–52.6) | 0.1466 |
| 35–39 | 50.2(43.8–56.4) | 45.2(38.3–52.3) | 55.0(51.5–58.5) | 0.1968 |
| 40–44 | 49.1(39.1–59.2) | 51.4(40.6–61.9) | 47.3(41.3–53.4) | 0.7643 |
| 45–49 | 44.6(22.5–69.1) | 46.9(24.4–70.7) | 42.5(28.1–58.3) | 0.8860 |
| **Marital Status** | | | | |
| Never Married | 55.2(48.0–62.2) | 52.7(46.1–59.2) | 54.2(50.4–57.9) | 0.8080 |
| Married/living together | 53.9(51.7–56.0) | 49.1(46.8–51.4) | 51.4(50.2–52.7) | 0.0510 |
| Divorced/separated/widowed | 58.2(53.2–65.4) | 56.8(49.0–64.3) | 46.6(42.4–50.7) | **0.0077** |
| **Religion** | | | | |
| Roman catholic | 55.6(51.6–59.5) | 53.3(48.8–57.9) | 51.0(48.3–53.6) | 0.0602 |
| Protestant/other Christian | 54.2(51.6–56.7) | 49.1(46.6–51.7) | 51.5(50.1–52.9) | 0.0660 |
| Muslim | 51.9(45.1–58.6) | 52.9(45.7–59.7) | 50.3(46.3–54.2) | 0.6887 |
| No religion | 55.9(43.7–67.4) | 42.9(32.1–54.4) | 48.2(41.5–55.0) | 0.2754 |
| **Birth Order** | | | | |
| 1 | 52.2(48.1–56.3) | 46.4(42.2–50.7) | 48.4(46.2–50.6) | 0.1099 |
| 2–4 | 55.9(53.0–58.8) | 51.1(48.2–54.1) | 51.0(49.5–52.6) | **0.0035** |
| 5 + | 53.3(49.4–57.1) | 51.2(47.2–55.1) | 55.5(53.1–57.9) | 0.3395 |
| **Parity** | | | | |
| 0–1 | 52.3(48.2–56.3) | 47.0(42.8–51.2) | 49.2(47.0–51.4) | 0.1867 |
| 2–4 | 56.5(53.7–59.2) | 51.5(48.6–54.3) | 50.7(49.2–52.3) | **0.0004** |
| 5 + | 51.6(47.5–55.7) | 49.9(45.6–54.2) | 55.8(53.2–58.3) | 0.0900 |
| **Number of children in Household** | | | | |
| 0–1 | 51.7(50.6–54.9) | 42.1(38.6–45.6) | 46.7(44.9–48.4) | **0.0097** |
| 2–4 | 55.8(53.3–58.3) | 54.0(51.4–56.6) | 54.6(53.1–56.1) | 0.4211 |
| 5 + | 56.6(34.8–76.1) | 58.5(43.4–72.1) | 55.8(43.0–68.0) | 0.9494 |
| **Education** | | | | |
| No Education | 52.0(46.8–57.1) | 54.5(48.6–60.3) | 57.2(53.9–60.5) | 0.0963 |
| Primary Incomplete | 56.6(53.3–59.9) | 51.5(47.9–55.1) | 54.5(52.3–56.6) | 0.2968 |
| Primary Complete | 54.3(50.5–58.1) | 48.7(45.0–52.5) | 50.1(47.9–52.3) | 0.0619 |
| Secondary + | 51.8(47.4–56.2) | 47.5(43.3–51.7) | 47.7(45.7–49.6) | 0.0955 |
| **Wealth Quintile** | | | | |
| Lowest | 52.0(47.9–56.0) | 52.6(48.4–56.8) | 57.1(54.8–59.3) | **0.0299** |
| Second | 59.1(57.4–63.4) | 55.2(50.5–59.8) | 52.1(49.5–54.7) | **0.0070** |
| Middle | 50.0(45.3–54.6) | 49.1(44.2–54.0) | 50.2(47.5–52.9) | 0.9418 |
| Fourth | 56.1(51.2–60.8) | 49.3(44.4–54.0) | 47.8(45.0–50.5) | **0.0036** |
| Highest | 54.7(50.1–59.3) | 43.1(38.5–47.8) | 47.2(44.7–49.8) | **0.0056** |

*(Continued)*

**Table 2.** (Continued)

| | 2003 (n = 2380) | 2008/09 (n = 2237) | 2014 (n = 7380) | P-value[a] |
|---|---|---|---|---|
| | % (95% CI) | % (95% CI) | % (95% CI) | |
| **Occupation** | | | | |
| Unemployed | 55.5(52.2–58.7) | 50.7(47.6–53.9) | 52.5(49.8–55.2) | 0.1630 |
| Employed | 53.6(53.6–56.1) | 49.5(46.7–52.2) | 50.1(48.0–52.1) | **0.0372** |
| **Province** | | | | |
| Nairobi | 57.0(48.8–64.8) | 52.3(43.6–60.9) | 49.6(46.0–53.2) | 0.1021 |
| Central | 47.4(41.3–53.7) | 46.0(38.4–53.8) | 43.4(39.7–47.2) | 0.2775 |
| Coast | 53.4(46.6–60.0) | 52.7(45.9–59.3) | 52.7(49.2–56.1) | 0.8561 |
| Eastern | 55.2(50.1–60.1) | 51.0(45.6–56.4) | 50.7(47.4–54.0) | 0.1426 |
| Nyanza | 56.2(51.0–61.3) | 58.3(53.6–62.8) | 52.2(49.1–55.2) | 0.1923 |
| Rift Valley | 55.6(51.8–59.4) | 42.9(39.2–46.7) | 54.3(52.2–56.4) | 0.5547 |
| Western | 56.6(56.1–62.0) | 51.9(45.7–58.0) | 49.9(46.5–55.2) | **0.0441** |
| North Eastern | 36.5(25.5–49.1) | 49.7(37.6–61.8) | 50.6(50.7–56.9) | **0.0468** |
| **Place of delivery** | | | | |
| Home | 55.4(52.8–58.0) | 50.7(47.9–53.5) | 52.8(50.8–54.7) | 0.1160 |
| Public | 51.1(47.0–55.1) | 48.1(44.5–51.6) | 52.1(50.5–53.7) | 0.6559 |
| Private | 54.9(49.5–60.2) | 51.6(45.3–57.8) | 45.2(42.4–48.1) | **0.0011** |
| **Total** | 54.3(54.9–56.3) | 50.0(48.0–52.1) | 51.3(50.2–52.4) | **0.0109** |

KDHS = Kenya Demographic Health Survey.

[a] P-values were calculated form the Cochrane-Armitage trend test.

children who were delivered in private sectors decreased from 54.9% (2003) to 51.6% (2008/09) and to 45.2% (2014); $P = 0.0011$.

## Trends of missed opportunities for vaccination (MOV) among children aged 0–23 months in Kenya 2003 to 2014

The results presented in Table 3 indicate a statistically significant increase in the percentage of children who experienced MOV from 22.7% in 2003 to 31.9% in 2008/09 and further to 37.6% in 2014 (P<0.0001). This trend was observed across all significant factors except for those residing in urban areas (24.5% in 2003 to 24.4% in 2008/09 to 32.5% in 2014; P = 0.0008), children with mothers who were Muslims (19.3% in 2003 to 35.5% in 2008/09 to 28.5% in 2014; P = 0.0088), those who had no religion (27.5% in 2003 to 35.2% in 2008/09 to 42.2% in 2014; P = 0.0325), children who belonged to the highest wealth quintile (24.8% in 2003 to 23.4% in 2008/09 to 31% in 2014; P = 0.0120), and those living in Nyanza Province (34.3% in 2008 to 39.9% in 2008/09 to 42.1% in 2014; P = 0.0096).

Fig 1 illustrates the trends non-vaccination, under-vaccination and missed opportunities for vaccination among children aged 0–23 months in Kenya; KDHS 2003, 2008/09 and 2014.

## Discussion

The main objective of this study was to assess the trends of non-vaccination, under-vaccination, and missed opportunities of vaccination among children aged 0–23 months in Kenya from 2003 to 2014. The study has established that over the 10-year period, there was a consistent decline in the trend of non-vaccination among children aged 0–23 months in Kenya, across most of the study variables except for women who were divorced/ separated or widowed

**Table 3. Trends of missed opportunities for vaccination among children aged 0–23 months in Kenya; KDHS 2003, 2008/09 and 2014.**

| | 2003 (n = 2380) | 2008/09 (n = 2237) | 2014 (n = 7380) | P-value[a] |
|---|---|---|---|---|
| | % (95% CI) | % (95% CI) | % (95% CI) | |
| **Sex of child** | | | | |
| Male | 23.9(21.5–26.4) | 33.3(30.6–36.0) | 39.1(37.5–40.7) | <**0.0001** |
| Female | 21.5(19.3–23.9) | 30.4(27.7–33.2) | 36.0(34.5–37.6) | <**0.0001** |
| **Residence** | | | | |
| Urban | 24.5(20.7–28.7) | 24.4(20.6–28.6) | 32.5(30.7–34.3) | **0.0008** |
| Rural | 22.3(20.5–24.2) | 33.8(31.6–36.0) | 40.4(39.0–41.8) | <**0.0001** |
| **Mother's Age** | | | | |
| 15–19 | 26.3(21.5–31.7) | 31.0(25.1–37.6) | 41.7(38.1–45.4) | <**0.0001** |
| 20–24 | 23.8(20.9–27.0) | 34.5(31.1–38.1) | 39.0(37.0–41.1) | <**0.0001** |
| 25–29 | 22.2(19.0–25.7) | 29.4(25.8–33.2) | 36.9(34.9–39.0) | <**0.0001** |
| 30–34 | 22.3(18.6–26.6) | 31.7(27.4–36.3) | 35.4(32.9–38.1) | <**0.0001** |
| 35–39 | 19.7(15.1–25.3) | 29.5(23.5–36.3) | 34.8(31.5–38.2) | <**0.0001** |
| 40–44 | 15.2(9.2–24.0) | 36.4(26.8–47.3) | 39.7(34.0–45.8) | <**0.0001** |
| 45–49 | 20.0(6.6–47.1) | 31.1(13.2–57.1) | 32.3(19.5–48.3) | 0.3584 |
| **Marital Status** | | | | |
| Never Married | 21.6(16.3–28.1) | 22.9(17.8–28.9) | 36.7(33.1–40.4) | **0.0001** |
| Married/ living together | 23.1(21.3–25.0) | 32.5(30.4–34.7) | 37.9(36.7–39.1) | <**0.0001** |
| Divorced/ separated/ widowed | 19.3(14.1–25.9) | 37.2(30.0–44.9) | 35.0(31.2–39.1) | **0.0001** |
| **Religion** | | | | |
| Roman catholic | 24.8(21.5–28.4) | 31.2(27.1–35.6) | 36.3(33.8–38.9) | <**0.0001** |
| Protestant/ other Christian | 21.8(19.8–24.0) | 31.3(29.0–33.7) | 38.9(37.6–40.2) | <**0.0001** |
| Muslim | 19.3(14.5–25.3) | 35.5(29.1–42.4) | 28.5(26.4–32.2) | **0.0088** |
| No religion | 27.5(18.0–39.5) | 35.2(25.2–46.7) | 42.2(41.1–49.0) | **0.0325** |
| **Birth Order** | | | | |
| 1 | 24.6(21.2–28.3) | 27.7(24.0–31.7) | 39.8(37.6–42.0) | <**0.0001** |
| 2–4 | 23.2(20.9–25.8) | 29.2(26.6–31.9) | 35.6(34.0–37.1) | <**0.0001** |
| 5 + | 20.1(17.2–23.3) | 40.6(36.7–44.5) | 39.7(37.3–42.1) | <**0.0001** |
| **Parity** | | | | |
| 0–1 | 25.4(22.1–29.1) | 27.9(24.2–31.8) | 40.2(38.0–42.3) | <**0.0001** |
| 2–4 | 22.3(20.0–24.7) | 30.6(28.0–33.3) | 35.6(34.1–37.1) | <**0.0001** |
| 5 + | 20.7(17.5–24.2) | 39.0(34.8–43.2) | 39.5(37.0–42.0) | <**0.0001** |
| **Number of children in Household** | | | | |
| 0–1 | 25.0(22.2–28.0) | 25.9(22.9–29.1) | 36.1(34.4–37.8) | <**0.0001** |
| 2–4 | 21.4(19.4–23.6) | 34.2(31.8–36.7) | 38.8(37.3–40.2) | <**0.0001** |
| 5 + | 18.5(6.8–41.5) | 62.4(47.3–75.5) | 31.5(20.9–44.4) | 0.2550 |
| **Education** | | | | |
| No Education | 18.3(14.6–22.7) | 43.6(37.9–49.6) | 33.2(30.1–36.4) | <**0.0001** |
| Primary Incomplete | 25.3(22.6–28.3) | 35.5(32.1–39.0) | 41.4(39.3–43.6) | <**0.0001** |
| Primary Complete | 21.5(18.5–24.8) | 30.3(27.0–33.8) | 36.6(34.5–38.7) | <**0.0001** |
| Secondary + | 22.8(19.3–26.7) | 23.2(19.9–27.0) | 36.7(34.8–38.6) | <**0.0001** |
| **Wealth Quintile** | | | | |
| Lowest | 21.5(18.4–25.1) | 38.5(34.5–42.7) | 39.1(36.9–41.4) | <**0.0001** |
| Second | 26.3(22.6–30.4) | 35.2(30.8–39.7) | 41.8(39.2–44.3) | <**0.0001** |
| Middle | 20.2(16.7–24.2) | 32.3(27.8–37.0) | 38.6(36.1–41.3) | <**0.0001** |
| Fourth | 20.3(16.6–24.5) | 28.3(24.2–32.8) | 37.0(34.4–39.7) | <**0.0001** |
| Highest | 24.8(21.0–29.0) | 23.4(19.7–27.6) | 31.0(30.1–33.4) | **0.0120** |

*(Continued)*

**Table 3.** (Continued)

| | 2003 (n = 2380) | 2008/09 (n = 2237) | 2014 (n = 7380) | P-value[a] |
|---|---|---|---|---|
| | % (95% CI) | % (95% CI) | % (95% CI) | |
| **Occupation** | | | | |
| Unemployed | 19.5(17.0–22.2) | 32.5(29.6–35.5) | 40.6(37.9–43.3) | <**0.0001** |
| Employed | 24.7(22.6–27.0) | 31.5(29.0–34.1) | 45.9(43.8–48.0) | <**0.0001** |
| **Province** | | | | |
| Nairobi | 22.8(16.7–30.3) | 30.8(23.3–39.4) | 27.9(24.8–31.2) | 0.2049 |
| Central | 16.2(12.1–21.3) | 19.7(14.2–26.7) | 33.2(29.7–36.8) | <**0.0001** |
| Coast | 18.1(13.5–23.9) | 37.4(31.1–44.1) | 37.5(34.3–40.9) | <**0.0001** |
| Eastern | 17.3(13.8–21.4) | 21.7(17.5–26.4) | 38.9(35.7–42.1) | <**0.0001** |
| Nyanza | 34.3(29.5–39.4) | 39.9(22.9–29.7) | 42.1(41.1–45.2) | **0.0096** |
| Rift Valley | 20.2(17.3–23.4) | 26.2(42.7–55.2) | 38.1(36.1–40.2) | <**0.0001** |
| Western | 32.0(27.0–37.4) | 49.0(42.9–55.2) | 46.5(43.1–49.8) | <**0.0001** |
| North Eastern | 10.9(5.2–21.4) | 35.0(24.3–47.4) | 19.2(14.6–24.8) | 0.1234 |
| **Place of delivery** | | | | |
| Home | 18.7(16.8–20.8) | 39.0(36.2–41.8) | 34.6(32.5–36.2) | <**0.0001** |
| Public | 26.9(23.4–30.7) | 22.9(20.0–26.0) | 40.0(38.4–41.6) | <**0.0001** |
| Private | 34.1(29.2–39.4) | 26.9(21.7–32.8) | 37.3(34.5–40.1) | 0.0575 |
| **Total** | 22.7(21.0–24.4) | 31.9(30.0–33.9) | 37.6(36.5–38.7) | <**0.0001** |

KDHS = Kenya Demographic Health Survey.

[a] P-values were calculated form the Cochrane-Armitage trend test.

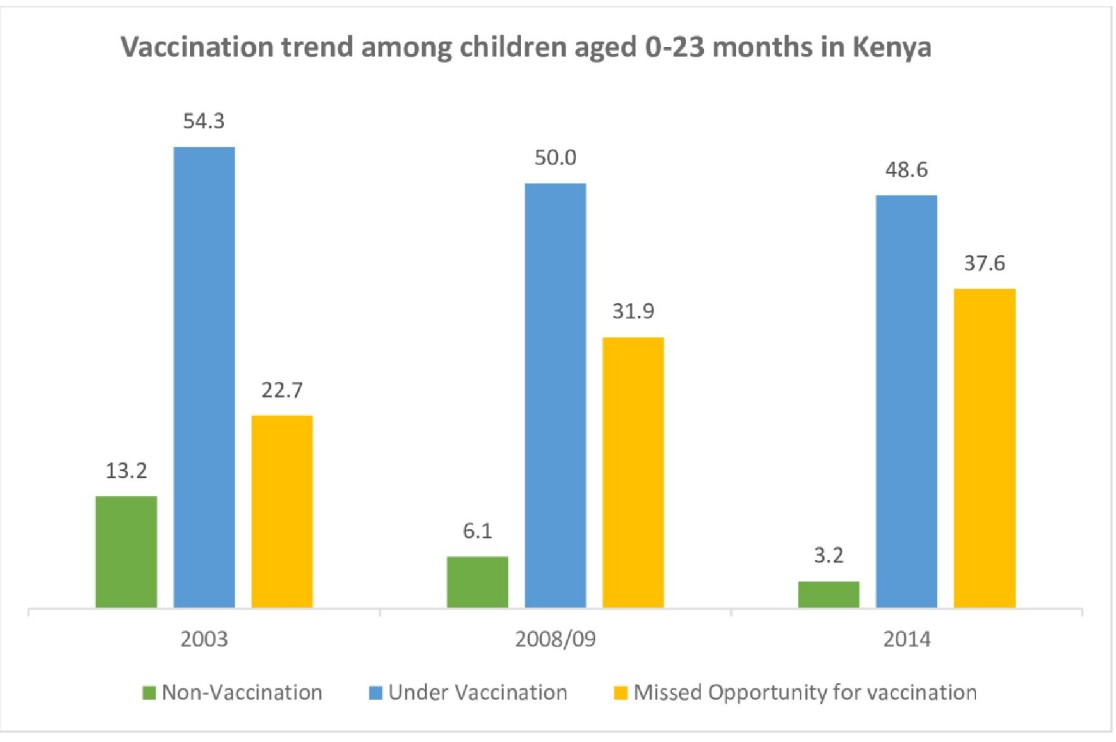

**Fig 1. Vaccination trends among children aged 0–23 months in Kenya.**

and for women residing in Nairobi Province. This consistent decline is aligned with the reported findings in the East Africa region and in Africa in general [12,18–20].

Non-vaccination is indicative of children within the immunization age not being in contact with immunization services and represents vulnerable group being left behind by multiple vaccination services and interventions [21]. The declining trend of non-vaccination in Kenya may be suggestive of improvement in immunization services in the country. These improvements may be attributed to increased advocacy, communication and social mobilization (ACSM) efforts by the immunization program and its partners [22].

As seen in other settings, these improvements in non-immunization trends could also possibly be explained by improvements in the maternal-child health programs in the country, increased investments in the health sector, increased investments in the vaccination and immunization programs in Kenya by the Ministry of health, immunization implementing partners and the County Governments [19,23–25].

The immunization program in Kenya is largely funded by the government of Kenya, supported by immunization partners such as Gavi, the Vaccine Alliance [26]. The flow of support (funding and technical) from donors to the Government of Kenya, for public health programs, is through the MOH, which in turn distributes them to the counties alongside other resources [27]. Assumedly, this provides a level ground in terms of resource allocation and distribution across the country. With this support, the Kenya MOH could have implemented immunization program activities that could have led to the observed improvements in terms of reductions in non-immunized children.

The reducing trend in non-vaccination across the variables in this study need to be sustained in order to have better immunization outcomes at the population and health system level.

There were varied outcomes in terms of levels and trends of under-vaccination among children aged 0–23 months in Kenya. Under-vaccination is indicative of the inability of the immunization system to attract and retain eligible children till they complete their immunization schedule. This inability could be due to several immunization system related factors. These factors could be related to availability, accessibility and affordability of these immunization services to the populations that need them most.

Over the study period, the levels of under-vaccination are lower than those seen around East Africa where about 69% of children received their required vaccines [20]. There were gender differences observed from this study with more males than females being vaccinated. However, different studies have reported a varying influence of gender on immunization with others reporting girls to be less likely to be vaccinated than boys [28]. In this study, more children in rural areas were vaccinated than those in urban areas. These differences in utilization of immunization services have been described before and may be due to proximity to service delivery points and possible higher travel costs [29].

A significant increasing trend was observed for those in the lowest wealth quintile and those living in NEP. Living in remote areas and urban poor populations have been documented as known inequity determinants for under-vaccination [19]. There is a significantly large number of refugees and nomadic populations amongst the inhabitants of the NEP [30]. Even within the NEP, there have been persistent disparities in vaccinations, with Somali children more likely to be vaccinated compared to non-Somalis [30].

A significant declining trend in under-vaccination was noticed in children of women aged 25–29, children of divorced/separated/widowed parents, children in 2–4 birth order and parity, second and fourth wealth quintiles, children living in the former Western Province and those born in private health facilities.

A significant mixed trend in under-vaccination was seen in families with one child at home, children living in the highest wealth quintiles where only close to half of their children were vaccinated. Studies in Kenya have shown full and timely vaccination to be higher amongst children of the rich and wealthy individuals with the hazard for being fully immunized being 10% more likely among children of the wealthiest compared to those of the poorest children [31]. A similar trend was also observed in the Democratic Republic of the Congo (DRC) where the proportion of fully immunized children were found to be higher in upper wealth quintiles [32]. A similar mixed trend in under-vaccination was seen amongst children with unemployed parents. However, a different study in Ethiopia using the DHS found the husband's employment status to significantly influence the full vaccination status of their children with the odds of full vaccination higher in children of mothers with employed husbands [33].

Over the 10-year period, the levels and trends of MOV among children aged 0–23 months in Kenya has generally increased across most variables. The MOV is indicative of the inability of the immunization system to provide children when they are free of contraindications their required immunization whenever they make contact with the health system [34]. In other words, the system is not able to utilize all opportunities to provide immunization services to deserving children [4]. This inability could be due to several factors such as health workers not checking vaccination status of children making contact with the health system, limited integration of vaccination services with other health services at the service delivery points, human resource shortages, poor retention of vaccination cards, and stock-outs of vaccines or related supplies [34].

A meta-analysis reviewing data from low-income countries [15], another study amongst the Maasai nomadic populations [35] and another one amongst children in a poor urban settlement of Nairobi, Kenya [36] found a MOV prevalence of 42%, 30% and 22%, respectively. The findings from this trend analysis has shown consistent significant rising trend and also aligns with regional trends of MOV in the region. However, there were non-significant trends of MOV that were observed amongst children with mothers aged 45–49, children in households with more than five children, those living in Nairobi and North-Eastern Provinces and those children born in private health facilities.

## Limitations

The quality of the secondary data used in this study is dependent on the quality of the data collection process. The accuracy and completeness of the data may be affected by errors in recording, data entry, and recall bias. Further, there may be missing or incomplete data on vaccination status, which may lead to inaccurate conclusions about vaccination trends. Similarly, the KDHS surveys may not have included all relevant variables that could impact vaccination rates, such as access to healthcare, socioeconomic status, and cultural beliefs. As the study is limited to data from 2003 to 2014, it therefore cannot capture more recent trends or changes in vaccination rates. Likewise, the study cannot establish a causal relationship between non-vaccination and other factors, as it is based on observational data and the findings may not be generalizable to other countries or regions with different healthcare systems, cultural beliefs, and socioeconomic conditions.

This study did not account for changes in the types and availability of vaccines over the study period, which could affect vaccination coverage and non-vaccination rates. Finally, the study only looked at non-vaccination trends, without considering the reasons for non-vaccination, which could limit the usefulness of the study in informing policy and interventions to improve vaccination coverage.

## Conclusion

The overall coverage trend of full vaccination was low among children aged 0–23 months in Kenya. Even though the benefits of most childhood vaccinations are scientifically unquestionable, vaccination coverage rates are far from 100% in many regions and show substantial variations.

The results show a significant decrease in the percentage of non-vaccinated children in Kenya from 2003 to 2014. However, some demographic groups, such as children in rural areas and those with higher birth orders or parities, still have higher non-vaccination rates than others. Overall, the results show that under-vaccination rates decreased in Kenya between 2003 and 2014. However, some groups, such as children in urban areas and those born to older mothers, still had high under-vaccination rates in 2014. Within the same study period, there was an increase in the proportion of missed opportunities for vaccination among children in Kenya, and there are significant differences in missed opportunities by various factors, including sex of child, residence, mother's age, marital status, religion, birth order, and parity.

## Recommendations

Targeted interventions need to be deployed to increase access to vaccination services and improve awareness in areas where children are not being vaccinated in Kenya. These interventions need to be tailored to target children in rural areas and those with higher birth orders or parities, women in urban areas such as Nairobi and those who are divorced/ separated or widowed.

In order to address the challenge of under-vaccination which has shown a varying trend across most study variables, the immunization program in Kenya needs to invest in immunization system related factors that would ensure children are retained in the immunization program for the period they are within the immunization schedule. These factors include those related to availability, accessibility and affordability and where possible, to ensure sustainability, strengthen their integration within routine childhood services offered in the same period.

This study results are indicative of the inability of Kenya's routine immunization system to provide children when they are free of contraindications their required immunization whenever they make contact with the health system. These findings highlight the need for targeted interventions to address the barriers to vaccination uptake in different subpopulations and settings to improve vaccination coverage and prevent vaccine-preventable diseases. Appropriate efforts are needed to ensure utilization of all opportunities to provide immunization services to deserving children. These include building the capacity of health workers to detect deserving children whenever they make contact with the health system, ensuring an un-interrupted vaccine supply chain and proper vaccination records documentation and retention practices.

More qualitative assessments with immunization services managers and mothers of children with immunization gaps are needed to have a good understanding of factors influencing trends in non-vaccination, under-vaccination and MOVs are required to provide full context for improving vaccination coverage and efforts at ensuring vaccination equity. The current study was not designed to provide this level of understanding.

## Acknowledgments

We appreciate and acknowledge the children and their caregivers for providing the KDHS data and the DHS program for the public availability of data that enabled this analysis.

## Author Contributions

**Conceptualization:** Christopher Ochieng' Odero.

**Data curation:** Christopher Ochieng' Odero.

**Formal analysis:** Christopher Ochieng' Odero, Vincent Omondi Were.

**Methodology:** Christopher Ochieng' Odero, Doreen Othero, Collins Ouma.

**Project administration:** Christopher Ochieng' Odero.

**Writing – original draft:** Christopher Ochieng' Odero.

**Writing – review & editing:** Christopher Ochieng' Odero, Doreen Othero, Vincent Omondi Were, Collins Ouma.

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
