## [Decision Letter · Decision Letter 0]

9 Aug 2023

PGPH-D-23-01014

Trends of non-vaccination, under-vaccination and missed opportunities for vaccination (2003-2014) amongst children 0-23 months in Kenya.

Dear Dr. Odero,

Thank you for submitting your manuscript to PLOS Global Public Health. After careful consideration, we feel that it has merit but does not fully meet PLOS Global Public Health’s publication criteria as it currently stands. Therefore, we invite you to submit a revised version of the manuscript that addresses the points raised during the review process.

The manuscript has been evaluated by two reviewers, and their comments are available below.

Both reviewers are very positive about your manuscript, but reviewer 1 has some recommendations for revisions. The reviewer requests additional information about how the weights were created to deal with the uneven distribution, and wonders if the 3 tables could be condensed. I will leave it to you to determine whether there is a way you can present the results more succinctly. Similarly, the reviewer would prefer the introduction to be more concise, but I do not think it overly long and will let you decide whether you wish to shorten it.

We look forward to receiving your revised manuscript.

Kind regards,

Steve Zimmerman, PhD

PLOS Staff Editor

Journal Requirements:

Additional Editor Comments (if provided):

Reviewers' comments:

Reviewer's Responses to Questions

**Comments to the Author**

1. Does this manuscript meet PLOS Global Public Health’s publication criteria? Is the manuscript technically sound, and do the data support the conclusions? The manuscript must describe methodologically and ethically rigorous research with conclusions that are appropriately drawn based on the data presented.

Reviewer #1: Yes

Reviewer #2: Yes

2. Has the statistical analysis been performed appropriately and rigorously?

Reviewer #1: Yes

Reviewer #2: Yes

3. Have the authors made all data underlying the findings in their manuscript fully available (please refer to the Data Availability Statement at the start of the manuscript PDF file)?

Reviewer #1: Yes

Reviewer #2: Yes

4. Is the manuscript presented in an intelligible fashion and written in standard English?

Reviewer #1: Yes

Reviewer #2: Yes

5. Review Comments to the Author

Reviewer #1: The introduction is long

would benefit from being more concise

the methodology is clear, but the explanation of how the unequal distribution is managed could be more detailed

In the results, the same table format is used 3 times, admittedly the data are different, but isn't it possible for each table to target the main message, given the length, or to indicate the major elements?

We appreciated the richness of the discussion, which covered studies from both southern and central Africa.

the limitations of the study are honest, especially in our context where cultural beliefs influence families' behaviour towards vaccination.

Reviewer #2: Dear Author(s)

Thanks for your efforts to provide a detailed description of the secondary data from the Kenya Demographic and Health Surveys (KDHS) conducted in 2003, 2008/9, and 2014. Your study design is quantitative, and it is good to have the opportunity to obtain the datasets, which are publicly available. In your manuscript, you describe the statistical methods used to analyze the data clearly, including weighted descriptive analysis and the Cochrane-Armitage trend test.

Your manuscript is well-organized and structured, with clear headings and subheadings that make it easy to navigate.

Since the latest data belong to 2014, your study reflects a cross-sectional image of trends of non-vaccination amongst children 0-23 months in Kenya about a decade ago, which you have open-heartedly expressed in "Limitations".

In conclusion, I think your study would be a fine contribution to the literature on vaccine hesitation and refusal.

6. PLOS authors have the option to publish the peer review history of their article (what does this mean?). If published, this will include your full peer review and any attached files.

**Do you want your identity to be public for this peer review?** For information about this choice, including consent withdrawal, please see our Privacy Policy.

Reviewer #1: **Yes: **NOUKEU NJINKUI Diomède

Reviewer #2: **Yes: **Selim ÖNCEL

---

## [Decision Letter · Decision Letter 1]

24 Nov 2023

PGPH-D-23-01014R1

Trends of non-vaccination, under-vaccination and missed opportunities for vaccination (2003-2014) amongst children 0-23 months in Kenya.

Dear Dr. Odero,

Thank you for submitting your manuscript to PLOS Global Public Health. After careful consideration, we feel that it has merit but does not fully meet PLOS Global Public Health’s publication criteria as it currently stands. Therefore, we invite you to submit a revised version of the manuscript that addresses the points raised during the review process.

Please note that we have only been able to secure a single reviewer to assess your revision. We are issuing a decision on your manuscript at this point to prevent further delays in the evaluation of your manuscript. Please be aware that the editor who handles your next revised manuscript might find it necessary to invite additional reviewers to assess this work once the revised manuscript is submitted. However, we will aim to proceed on the basis of this single review if possible.

We look forward to receiving your revised manuscript.

Kind regards,

Jianhong Zhou

Staff Editor

Journal Requirements:

Additional Editor Comments (if provided):

Reviewers' comments:

Reviewer's Responses to Questions

**Comments to the Author**

1. If the authors have adequately addressed your comments raised in a previous round of review and you feel that this manuscript is now acceptable for publication, you may indicate that here to bypass the “Comments to the Author” section, enter your conflict of interest statement in the “Confidential to Editor” section, and submit your "Accept" recommendation.

Reviewer #3: All comments have been addressed

2. Does this manuscript meet PLOS Global Public Health’s publication criteria? Is the manuscript technically sound, and do the data support the conclusions? The manuscript must describe methodologically and ethically rigorous research with conclusions that are appropriately drawn based on the data presented.

Reviewer #3: Partly

3. Has the statistical analysis been performed appropriately and rigorously?

Reviewer #3: No

4. Have the authors made all data underlying the findings in their manuscript fully available (please refer to the Data Availability Statement at the start of the manuscript PDF file)?

Reviewer #3: Yes

5. Is the manuscript presented in an intelligible fashion and written in standard English?

Reviewer #3: No

6. Review Comments to the Author

Reviewer #3: This study aimed to assess trends of non-vaccination, under-vaccination, and MOV among children aged 0-23 months using the DHS data in Kenya from 2003 to 2014. The manuscript is interesting, however, lacks some important information in the main text. Here are some points I would like the authors to consider to further highlight the contribution of the study.

1. Line 38, “51.3” should be “51.3%”?

2. The introduction section of the article should show the main research question and purpose. Currently it is unclear and too long.

3. Line 73-75, “In 1993….” should be formatted the English correctly. And the language of the entire article needs to be revised by native speaker further.

4. Line 159-168, Please show more details on the sample size calculation methods. What is the calculated formula and parameters you used?

5. What is the statistical method you used to arrive at the results for tables 1-3? Is it a regression? Because there are 3 columns of results for 2003-2014 but only one p value.

6. Table 1-3, All tables need to be presented in a three-line table that conforms to the format of the publication. And please double-check the numbers in all the tables, for example, Table 1 has a problem with the number 3.5 (6.5-16.8) for mother's aged 45-49, in 2014 column.

7. How can the results section adequately support the findings and conclusions of the study when there is no figure with the data presented? Please add some figures.

7. PLOS authors have the option to publish the peer review history of their article (what does this mean?). If published, this will include your full peer review and any attached files.

**Do you want your identity to be public for this peer review?** For information about this choice, including consent withdrawal, please see our Privacy Policy.

Reviewer #3: No

---

## [Editor Report · Decision Letter 2]

23 Jan 2024

Trends of non-vaccination, under-vaccination and missed opportunities for vaccination (2003-2014) amongst children 0-23 months in Kenya.

PGPH-D-23-01014R2

Dear Mr. Odero,

We are pleased to inform you that your manuscript 'Trends of non-vaccination, under-vaccination and missed opportunities for vaccination (2003-2014) amongst children 0-23 months in Kenya.' has been provisionally accepted for publication in PLOS Global Public Health.

Best regards,

Ramachandran Thiruvengadam, M.D.,

Academic Editor